# DTVF: A User-Friendly Tool for Virulence Factor Prediction Based on ProtT5 and Deep Transfer Learning Models

**DOI:** 10.3390/genes15091170

**Published:** 2024-09-05

**Authors:** Jiawei Sun, Hongbo Yin, Chenxiao Ju, Yongheng Wang, Zhiyuan Yang

**Affiliations:** 1School of Artificial Intelligence, Hangzhou Dianzi University, Hangzhou 310018, China; jwsun@hdu.edu.cn (J.S.); wyh520520520@163.com (Y.W.); 2School of Geography, University of Leeds, Leeds LS2 9JT, UK; ml22hy2@leeds.ac.uk; 3School of Electrical and Computer Engineering, University of Sydney, Camperdown, NSW 2006, Australia; chju4673@uni.sydney.edu.au

**Keywords:** deep learning, transfer learning, virulence factor, bioinformatics

## Abstract

Virulencefactors (VFs) are key molecules that enable pathogens to evade the immune systems of the host. These factors are crucial for revealing the pathogenic processes of microbes and drug discovery. Identification of virulence factors in microbes become an important problem in the field of bioinformatics. To address this problem, this study proposes a novel model DTVF (Deep Transfer Learning for Virulence Factor Prediction), which integrates the ProtT5 protein sequence extraction model with a dual-channel deep learning model. In the dual-channel deep learning model, we innovatively integrate long short-term memory (LSTM) with convolutional neural networks (CNNs), creating a novel integrated architecture. Furthermore, by incorporating the attention mechanism, the accuracy of VF detection was significantly enhanced. We evaluated the DTVF model against other excellent-performing models in the field. DTVF demonstrates superior performance, achieving an accuracy rate of 84.55% and an AUROC of 92.08% on the benchmark dataset. DTVF shows state-of-the-art performance in this field, surpassing the existing models in nearly all metrics. To facilitate the use of biologists, we have also developed an interactive web-based user interface version of DTVF based on Gradio.

## 1. Introduction

Virulence factors (VFs) are critical molecules in the infection process of the pathogen, leading to disease in the host. These factors impact the host through various mechanisms, such as promoting adhesion and invasion via membrane proteins or altering the host cell environment through secreted proteins like toxins [1]. Pathogens utilize distinct secretion systems to transfer proteins from the cytoplasm to the host or extracellular matrix, with Types I–IV secretion systems being the most common [2]. Additionally, some bacteria enhance their survival in environmental conditions and resist host immune responses by forming biofilms or producing siderophores, among other mechanisms.

Identifying and understanding these virulence factors are crucial for developing vaccines and novel therapeutics. By precisely locating these factors and studying their effects on the host, scientists can devise strategies to block these key interactions, thereby effectively preventing and treating diseases caused by pathogens [3]. Therefore, the identification of virulence factors not only aids in providing us with a deeper understanding of pathogenic mechanisms but also drives innovation in the research and treatment of infectious diseases.

Due to the importance of the problem, extensive research has been conducted in this area. In 2005, a strategic initiative focused on the specific category of virulence factors known as adhesins. A software application called SPAAN, utilizing neural networks, was developed to accurately predict these adhesins, achieving high precision in identifying them from a broad range of bacteria [4]. To facilitate user interaction, scientists have developed a web server named VirulentPred. This is a machine learning-based method that utilizes support vector machines (SVMs) to predict virulent proteins in bacterial pathogens. It employs a bi-layer cascade SVM architecture to analyze protein sequence features, providing a web server for broader applications in identifying virulence factors [5]. In 2012, Zheng et al. [6] proposed a novel network-based approach that integrates protein–protein interaction (PPI) data from the STRING database to enhance the identification process of virulence factors. This method demonstrated a significant improvement in accuracy compared to traditional sequence-based approaches. However, its effectiveness largely depends on the availability and accuracy of the PPI data. Subsequently, researchers introduced “MP3”, an innovative tool that employs a combined approach of an SVM and a Hidden Markov Model to predict pathogenic proteins in genomic and metagenomic datasets. MP3 demonstrated superior performance over VirulentPred across three distinct datasets [7]. In 2020, Rentzsch et al. [8] proposed an effective negative data selection strategy named PBVF to construct a novel and diversified dataset. Building on this foundation, they evaluated various classifiers based on an SVM and Random Forest, approaches based on direct sequence similarity, and their combinations for predicting bacterial virulence factors. They discovered that direct sequence similarity plays a crucial role in the identification of VFs and that integrating it with other features into machine learning models significantly enhances performance. Moreover, methods based on sequence similarity demonstrated superior performance compared to the MP3 method when using the same training dataset.

In recent years, with the rapid advancements in artificial intelligence technologies, the application of machine learning and deep learning to the identification of virulence factors has become a significant direction. In 2021, Xie et al. introduced DeepVF [9], a hybrid framework based on deep learning that utilizes a stacking strategy to identify VFs, showing higher accuracy compared to other predictive models. Recently, Singh et al. proposed VF-Pred [10], an innovative framework designed to detect virulence factors from genomic data. This framework significantly enhances prediction accuracy by incorporating a novel Seq-Alignment feature. Research results indicate that VF-Pred demonstrates exceptional performance, achieving an accuracy of 83.5%, surpassing existing methods used for VF detection.

In previous studies, extensive research has been conducted on virulence factors, leading to the development of various computational prediction models. The majority of these approaches have employed the Position-Specific Scoring Matrix, Dipeptide Composition, and other features based on physicochemical properties and protein sequence composition for feature engineering. Models have been constructed using machine learning algorithms, such as XGBoost and Random Forest, training them with different combinations of these features.

With the advancement of Natural Language Processing (NLP) technologies and increased GPU computational power, the feasibility of using large-scale pre-trained models for feature extraction and training predictors through transfer learning has been progressively validated. Based on these advancements, we presented a novel approach for identifying potential VFs called DTVF, which is a dual-channel deep learning model with an attention mechanism, utilizing the large-scale pre-trained transformer model ProtT5 as a feature extractor. Compared to traditional models, this approach enables the model to more effectively capture complex sequence patterns and contextual dependencies within protein sequences. The DTVF model integrates LSTM and CNN within its architecture and employs a transfer learning strategy, which not only enhances the adaptability of the model to diverse datasets but also facilitates its application to novel pathogens with limited data availability, thereby ensuring that DTVF achieves state-of-the-art performance across various benchmarks. The final trained DTVF model can be accessed via a web-based user interface. By uploading the embedding .h5 file, the interface can return the probability of it being VF. This web-based UI helps researchers quickly and efficiently screen large datasets to identify samples that are VFs, thereby saving a significant amount of time and resources.

## 2. Materials and Methods

### 2.1. Dataset

In this study, a pre-existing dataset derived from antecedent research [9] was accessed and deployed. The dataset comprised 9749 virulence factors (VFs) sourced from three public repositories (Victors [11], VFDB [12], and PATRIC [13]), which are pertinent to bacterial pathologies, with the objective of establishing an updated and exhaustive compendium.

In addition, a cohort of 66,982 non-VFs was meticulously selected from the VFDB by utilizing a sophisticated methodology for the procurement of negative data samples. To mitigate the issue of sequence redundancy, both the positive and negative datasets underwent a process of clustering. The clustering was performed using the CD-HIT program, which grouped similar sequences together based on a sequence identity threshold of 0.3, and redundant sequences were excluded by selecting representative sequences from each cluster to create a non-redundant dataset [14].

Consequently, the resultant non-redundant dataset comprised 3576 VFs and 4910 non-VFs. The distributions of sequence length in the training set and test set are shown in Figure 1.

### 2.2. Feature Extraction

The ProtT5 [15] feature extractor was employed to process all protein sequences, which is a pre-trained model that utilizes T5 architecture that has been trained on a dataset of protein sequences. The version of the pre-trained model utilized was designated as ProtT5-XL-BFD, which underwent pre-training on the BFD dataset. The BFD dataset comprises a collection of 2.1 billion protein sequences. This model encompasses a total of approximately three billion parameters. For each protein sequence of varying lengths, the feature extractor generated a feature vector of 1024 elements based on its sequence. These feature vectors, in conjunction with the corresponding labels, were subsequently input into the model as features.

### 2.3. Model Building

In this study, a dual-channel model was deployed to process the features of protein sequences, based on learning these characteristics, ultimately achieving the function of identifying potential virulence factors. This dual-channel model consisted of an LSTM module and a CNN module, with a dot-product self-attention layer additionally incorporated into each module separately. The conversion relationships between the layers of CNN module are shown in Appendix A. The structure of this model is shown in Figure 2.

The LSTM module is a multi-layered recurrent neural network (RNN). Owing to the uniform length of the input protein sequence features, which are vectors of 1024, the long short-term memory network (LSTM) was selected as the principal framework for this module to mitigate potential gradient vanishing issues associated with long sequences. We define the input sequence as $x=\left(x_{1},x_{2},\ldots ,x_{T}\right)\in R^{d_{x}}$, where x represents the protein-encoding vector passed into the input layer, and $d_{x}$ denotes the input dimensionality of the vector. Thus, the mathematical formulation for the LSTM layer can be expressed as
$$\begin{array}{cc} i_{t} & =\sigma (W_{i}x_{t}+U_{i}h_{t-1}+b_{i})\\ f_{t} & =\sigma (W_{f}x_{t}+U_{f}h_{t-1}+b_{f})\\ o_{t} & =\sigma (W_{o}x_{t}+U_{o}h_{t-1}+b_{o})\\ {\tilde{c}}_{t} & =\tanh(W_{c}x_{t}+U_{c}h_{t-1}+b_{c})\\ c_{t} & =i_{t}⊙{\tilde{c}}_{t}+f_{t}⊙c_{t-1}\\ h_{t} & =o_{t}⊙\tanh\left(c_{t}\right) \end{array}$$
where $W_{i}$, $W_{f}$, $W_{o}$, $W_{c}\in R^{d_{h}\times d_{x}}$, and $U_{i}$, $U_{f}$, $U_{o}$, $U_{c}\in R^{d_{h}\times d_{h}}$ are tunable hyperparameters of the LSTM layer, $d_{h}$ is the dimension of the hidden layer, $x_{t}$, $h_{t}$, and $c_{t}$, respectively, represent the input, hidden state, and cell state at time step *t*, respecitvely, and σ is the sigmoid function. Furthermore, we added a dropout layer both before and after the LSTM layer to solve the overfitting problem. On the other hand, to better capture some local information of the input feature vectors and enhance the generalization capability of the model, the CNN module was introduced in parallel as a part of the entire model. This component was trained concurrently with the LSTM module.

For the input vector $x=\left(x_{1},x_{2},\ldots ,x_{T}\right)\in R^{d_{x}}$, the convolutional operation of the CNN layer can be expressed as follows:$$\begin{array}{cc} y_{i} & =f(w\cdot x_{i-1:i+h-2}+b) \end{array}$$
where *h* is the window size (we set h=3 in this study), w is the convolution kernel, *b* is the bias term, f(·) is the activation function, and $x_{i-1:i+h}$ is a subsequence composed of the elements from position i−1 to i+h−2 within the input vector. The convolution operations were performed by traversing from left to right on the input vectors, yielding the feature representation of the input vector, as follows:$$\begin{array}{cc} Y & =[y_{1},y_{2},\cdots ,y_{d_{x}-h+1}] \end{array}$$

The input vectors’ size is 1024; padding, as is customary in typical convolutional operations, is not required. Consequently, Y represents the output of the convolutional layer.

Following the CNN layer, we have incorporated an Attention module and a batch normalization layer. Similar to the LSTM module, we have added a Dropout layer both before and after the CNN layer.

To enhance the capture of potential positional correlations among input features, self-attention mechanisms were incorporated subsequently to both CNN and LSTM modules, enabling a more precise prediction of the output at each position by focusing on the inter-relations of different segments within the sequence.

The inputs of the Attention layer are the outputs of the LSTM layer ${\mathrm{Out}}_{L}$ and the outputs of the CNN layer ${\mathrm{Out}}_{C}$. To enhance the expressive capacity of the model, we need a learnable linear transformation to process the input sequence. We apply linear transformations denoted as $w_{1}$ and $w_{2}$ to ${\mathrm{Out}}_{L}$ and ${\mathrm{Out}}_{C}$ to achieve this goal:$$\begin{array}{cc} {\mathrm{Out}}_{L}^{li} & =w_{1}\cdot {\mathrm{Out}}_{L}\\ {\mathrm{Out}}_{C}^{li} & =w_{2}\cdot {\mathrm{Out}}_{C} \end{array}$$

The self-attention scores were calculated for each element in the tensors ${\mathrm{Out}}_{L}^{li}$ and ${\mathrm{Out}}_{C}^{li}$, which involves performing batch matrix multiplications of each tensor with its transpose. The self-attention score tensors were normalized through the Softmax function to obtain the self-attention weight matrices, as follows:$$\begin{array}{cc} {\mathrm{Attn}}_{L} & =\mathrm{softmax}({\mathrm{Out}}_{L}^{li}\cdot {\left({\mathrm{Out}}_{L}^{li}\right)}^{T})\\ {\mathrm{Attn}}_{C} & =\mathrm{softmax}({\mathrm{Out}}_{C}^{li}\cdot {\left({\mathrm{Out}}_{C}^{li}\right)}^{T}) \end{array}$$

Ultimately, the outputs of the LSTM and CNN components were multiplied by the corresponding self-attention weight matrices, resulting in the final outputs of the LSTM module and CNN module. These outputs were then balanced through a weighted sum node to obtain the final output of the DTVF model. The conversion relationships between the layers of DTVF model are shown in Appendix A.

### 2.4. Hyperparameter Search

In pursuit of selecting the optimal hyperparameters for tuning the model, six hyperparameters were established. These parameters were designated to adjust the hidden layer size and the probability of the dropout layer for both the CNN block and the LSTM block, the number of layers in the LSTM block, and the learning rate of the model. The method of 10-fold cross-validation was employed in the hyperparameter search process, with Optuna [16] being utilized for this procedure. The optimal parameters were discovered and are presented in the Appendix A. The DTVF model was trained on a computing platform equipped with two NVIDIA A10 GPUs. The search range of hyperparameters is shown in Table 1.

### 2.5. Web-Based UI

We developed an interactive web-based user interface (UI) using Gradio, which facilitates the upload of embeddings (fixed length of 1024) extracted by the pre-trained ProtT5 model in .h5 format. This service returns the virulence factor probability for these protein sequences. Additionally, the UI supports batch uploads and, in such cases, displays a pie chart depicting the proportion of virulence factors (VF) within the dataset.

The primary advantages of our solution are as follows:Intuitive Interface: The web-based UI is designed to be accessible and user-friendly, accommodating users with varying levels of technical expertise. This ensures that a wide range of researchers can utilize the tool effectively.Real-Time Data Processing: The interactive components of Gradio enable users to upload and process data in real time. This functionality provides prompt results for both individual samples and batch data.Data Visualization: For batch uploads, the UI not only returns prediction results but also generates a pie chart that visually represents the proportion of VF in the dataset. This visualization capability enhances data analysis and supports informed decision making.

## 3. Results

### 3.1. Model Performance Evaluation

We evaluated the performance of our model using some metrics such as accuracy (ACC), sensitivity (SN), precision (PR), F1-Score (FS), specificity (SP), and the area under the ROC curve (AUROC). They are calculated as follows:$$\begin{array}{cc} \mathrm{ACC} & =\frac{\mathrm{TP}+\mathrm{TN}}{\mathrm{TP}+\mathrm{FP}+\mathrm{TN}+\mathrm{FN}}\\ \mathrm{SN} & =\frac{\mathrm{TP}}{\mathrm{TP}+\mathrm{FN}}\\ \mathrm{PR} & =\frac{\mathrm{TP}}{\mathrm{TP}+\mathrm{FP}}\\ \mathrm{FS} & =\frac{2\times \mathrm{PR}\times \mathrm{SN}}{\mathrm{PR}+\mathrm{SN}}\\ \mathrm{SP} & =\frac{\mathrm{TN}}{\mathrm{TN}+\mathrm{FP}} \end{array}$$
where TP represents the true positives, TN represents the true negatives, FP represents the false positives, and FN represents the false negatives.

### 3.2. Ablation Study

To ascertain the influence of various network architectures and attention mechanisms on the task of predicting virulence factors, we conducted an investigation using six distinct models trained on an identical dataset. The performance of these models was rigorously evaluated against a consistent, independent test set. The outcomes of this ablation study, detailed in Table 2, reveal that models incorporating attention mechanisms significantly outperformed those without such mechanisms across both long short-term memory (LSTM) and convolutional neural network (CNN) architectures.

As the result shows in Table 2, the BiLSTM model achieved an accuracy of 0.8220, sensitivity of 0.7205, precision of 0.9041, F1-Score of 0.8019, specificity of 0.9236, and an area under the receiver operating characteristic curve of 0.9124. In comparison, the CNN model yielded slightly lower performance with an ACC of 0.8038, SN of 0.6979, PR of 0.8855, FS of 0.7806, SP of 0.9097, and AUROC of 0.8915.

Introducing attention mechanisms in the CNN-Att model resulted in improved performance, with an ACC of 0.8168, SN of 0.7882, PR of 0.8361, FS of 0.8114, SP of 0.8455, and AUROC of 0.8941. Further enhancements were observed with the CNN-Multi model, which achieved an ACC of 0.8281, SN of 0.7830, PR of 0.8607, FS of 0.8200, SP of 0.8732, and AUROC of 0.9101.

The DualModel (DTVF), our proposed model, demonstrated superior performance across all metrics, with an ACC of 0.8455, SN of 0.8021, PR of 0.8783, FS of 0.8385, SP of 0.8889, and AUROC of 0.9208. These findings substantiate the effectiveness of attention mechanisms within this specific context.

Moreover, the hybrid model, which amalgamates LSTM and CNN networks augmented with attention mechanisms, exhibited superior performance, surpassing all competing models across every evaluated metric. This underscores the robustness and efficacy of the proposed DualModel (DTVF) for predicting virulence factors.

### 3.3. Performance of DTVF on Independent Test Set

Upon the completion of the training phase, wherein the DTVF model was conditioned using the training set and meticulously selected hyperparameters, the model was subsequently deployed to perform inference on an independent test set. A comparative analysis was conducted, juxtaposing the predictive outcomes with the actual labels. Following this analysis, the performance metrics of the DTVF model were elucidated through the construction of a receiver operating characteristic (ROC) curve, a precision-recall (PR) curve, and a confusion matrix.

To provide a multidimensional assessment of the model performance, a radar chart was created, offering a holistic depiction of the model efficacy across various metrics. This comprehensive methodological approach enabled a more detailed evaluation of predictive capabilities of the DTVF model. The effectiveness of the DTVF model, as demonstrated by the independent test set, is illustrated in Figure 3, Figure 4 and Figure 5.

### 3.4. Performance Comparison with Existing Models

We compared the performance of our model to other VF predictors, such as VirulentPred [5], MP3 [7], PBVF [8], DeepVF [9], and VF-Pred [10]. Because our dataset was previously evaluated by these tools, their evaluation results were directly taken from published studies. The results are presented in Table 3. In the comparative table, the precision metric was omitted due to its absence from previous studies. It is observable that across the four metrics of accuracy, F1-Score, specificity, and AUROC, the DTVF model surpasses the most recent models. In comparison to the latest VF-Pred model released in 2024, the DTVF model exhibits a 1% increase in accuracy, a 3.89% enhancement in specificity, and a significant 8.57% improvement in the AUROC, which is a key indicator for classification performance.

### 3.5. Web-Based UI Workflow and Results

The proposed demonstration workflow encompasses several critical steps, utilizing advanced machine learning models and user-friendly interfaces to assess the potential toxicity of protein sequences. Initially, users input the protein sequences of interest, which are subsequently processed through the Prot-T5 model to generate embeddings—the vector representations of these protein sequences. These embeddings, stored in the .h5 format, are then uploaded to the user interface provided. This embedding operation is shown in a dynamic demo image in Appendix A.

Upon receiving the embeddings, the pre-trained DTVF model conducts the inference process. This model evaluates each embedding to determine the likelihood that the corresponding protein sequence exhibits characteristics of a potential virulence factor. The results of this analysis are quantified as scores, which are displayed on the frontend. This prediction operation is shown in a dynamic demo image in Appendix A.

A distinctive feature of this workflow is its ability to efficiently manage batch operations. This allows users to upload multiple embeddings simultaneously, thereby enhancing throughput and utility of the system. Following the analysis, the system generates a pie chart that visually represents the proportion of protein sequences identified as potential virulence factors within the batch. This graphical representation facilitates the intuitive interpretation of the results, offering a clear overview of the dataset composition in terms of virulence possibility.

This workflow integrates sophisticated computational techniques with a streamlined user interface, enabling the rapid and accurate assessment of protein sequence toxicity. This, in turn, supports research and decision-making processes in the fields of bioinformatics and molecular biology. The pipeline of the workflow of the DTVF user interface is shown in Figure 6.

## 4. Discussions

### 4.1. Significance and Future Directions

The DTVF model proposed in this paper has demonstrated exceptional performance, showcasing a significant practical application value in the fields of microbiology and biomedicine, particularly in responding to the urgent situations of infectious disease outbreaks [12]. This model efficiently identifies virulence factors in emerging pathogens, providing invaluable guidance for the design of targeted therapies and the rapid development of vaccines. Consequently, it directly enhances the precision and timeliness of medical interventions.

Moreover, the WebUI interface constructed in this study substantially lowers the technical barriers, enabling non-professional medical personnel to easily handle and analyze data efficiently. This innovation not only simplifies operational procedures but also significantly improves work efficiency and the readability of results through batch uploads and data visualization functions. These features offer more intuitive and clear foundations for medical decision making.

In this study, we plan to deepen and expand the DTVF model from multiple dimensions. We also intend to broaden the scope of model training to include more diverse data types, especially incorporating structural information and functional annotations [17]. This expansion aims to improve the accuracy and comprehensiveness of virulence factor prediction. Such an initiative is expected to unveil more secrets of pathogen virulence mechanisms, paving the way for novel approaches to disease prevention and treatment.

Simultaneously, considering the potential noise or adversarial interference [18] in practical applications, we will actively explore integrating adversarial training techniques into the DTVF model. This integration aims to enhance the model’s robustness and stability [19], ensuring it maintains high performance in complex and variable real-world environments.

### 4.2. Limitations and Challenges

Despite the DTVF model demonstrating efficient and accurate results, several limitations and challenges remain. The high dependency of the model on training data quality and diversity is a critical issue. The performance of the DTVF model largely relies on the availability of well-annotated virulence factor datasets. However, the currently available datasets are limited in number and are difficult to obtain. The scarcity of these data restricts the model’s potential for broader applications across various pathogens and more complex environments.

The common challenge of interpretability [20] in deep learning models also affects the further application of the DTVF model. Although the model can provide highly accurate predictions, its internal mechanisms and decision-making processes are often difficult to explain and understand. This “black box” nature may hinder researchers’ understanding of underlying biological mechanisms, thereby limiting the broad acceptance in biomedical research and clinical applications.

To overcome these limitations and challenges, future research must focus on constructing larger-scale and higher-quality virulence factor datasets to support continuous model optimization and generalization capability enhancement. Additionally, exploring new methods and techniques to improve the interpretability of deep learning models, such as feature importance analysis and visualization techniques [21], is crucial. These efforts will help us gain a more comprehensive understanding of the predictive mechanisms and promote the model’s widespread application in microbiology and biomedicine.

## 5. Conclusions

In this study, we introduced DTVF, a novel dual-channel deep learning model integrated with an attention mechanism for the identification of VFs. By leveraging the advanced ProtT5 feature extractor and combining the strengths of LSTM and CNN architectures, DTVF achieved state-of-the-art performance in the benchmark. Experimental results demonstrated that the model outperforms existing methods in terms of accuracy, sensitivity, and specificity, underscoring its robustness and effectiveness in predicting virulence factors from protein sequences.

Beyond its technical advancements, DTVF holds significant practical implications for the fields of microbiology and biomedicine, particularly in the rapid identification of VFs in emerging pathogens. This capability is crucial for the timely development of targeted therapies and vaccines, which are essential during infectious disease outbreaks.

## Figures and Tables

**Figure 1 genes-15-01170-f001:**
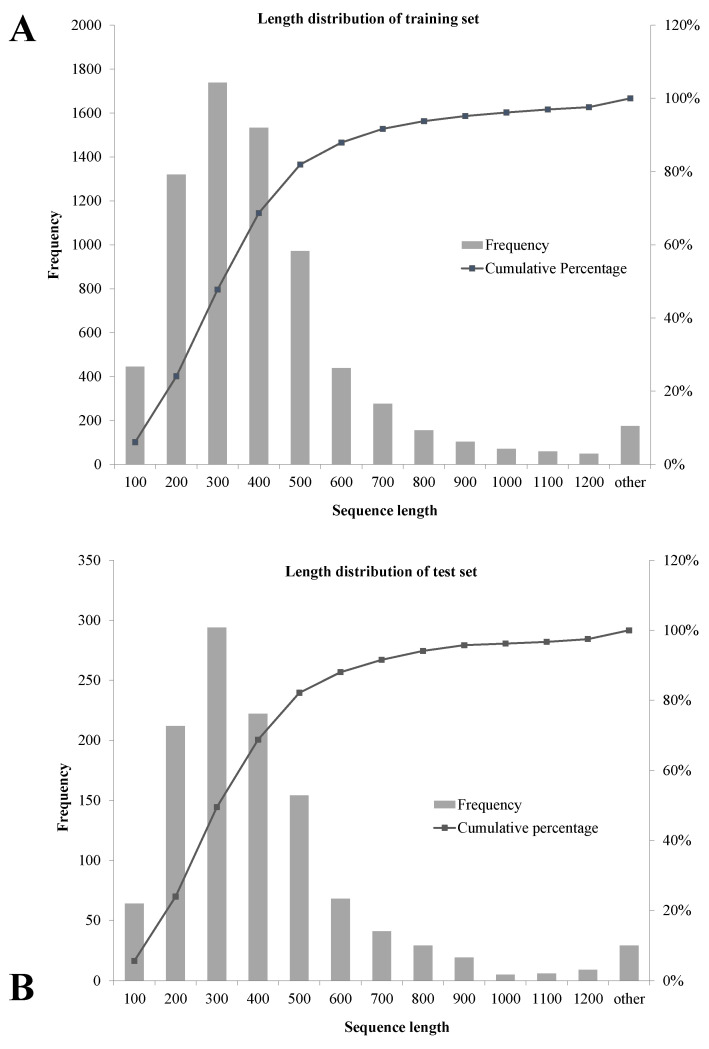
Distribution of sequence length for the training dataset and test dataset. (**A**) Training set. (**B**) Test set.

**Figure 2 genes-15-01170-f002:**
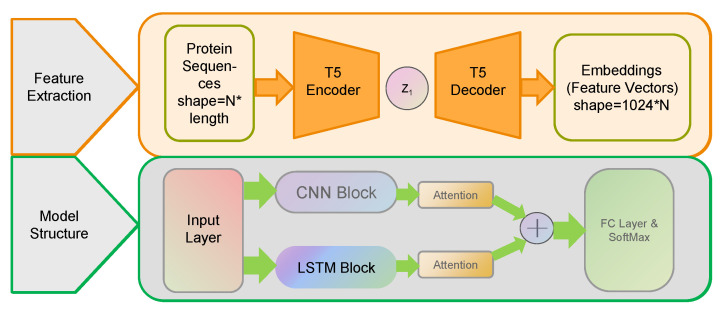
Framework of DTVF model. This framework included two models: feature extraction and prediction.

**Figure 3 genes-15-01170-f003:**
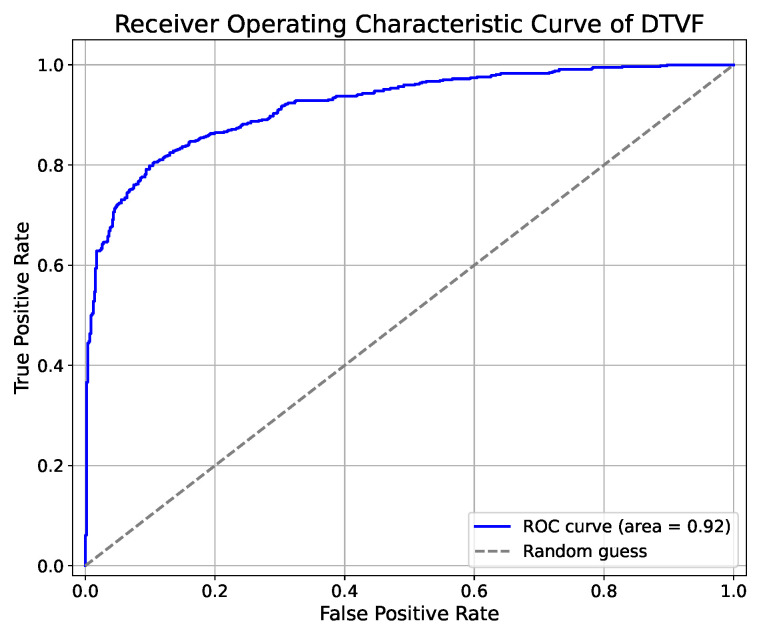
ROC curve of DTVF model on independent test set.

**Figure 4 genes-15-01170-f004:**
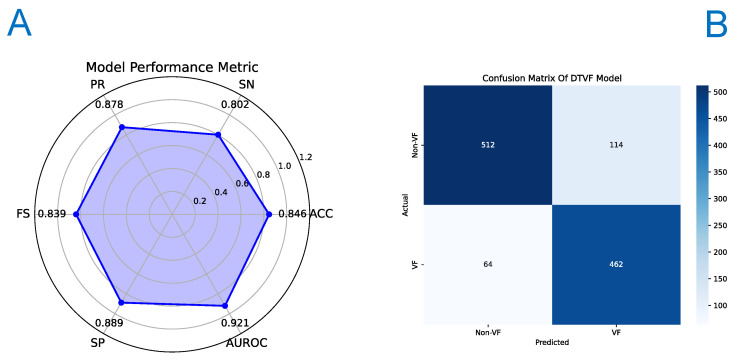
Radar graph and confusion matrix of DTVF model on independent test set. (**A**) Radar graph. (**B**) Confusion matrix.

**Figure 5 genes-15-01170-f005:**
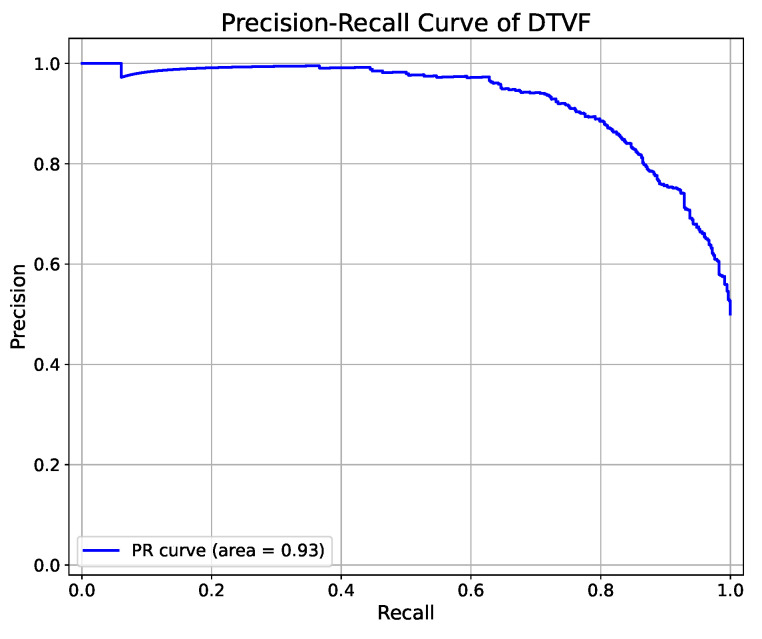
PR curve of DTVF model on independent test set.

**Figure 6 genes-15-01170-f006:**
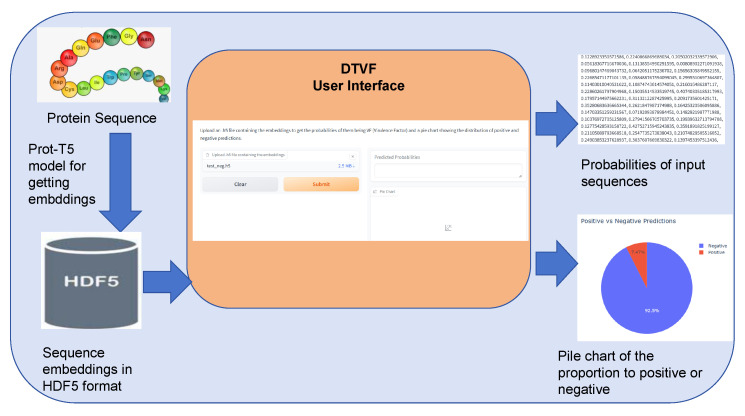
Pipeline of the workflow of DTVF user interface.

**Table 1 genes-15-01170-t001:** Hyperparameters search range of DTVF model.

Model	Hyperparameter Search Range	Chosen Parameters
BiLSTM	Hidden size: [64, 512]	Hidden size: 482
	Num layers: [2, 5]	Num layers: 2
	Dropout ratio: [0.1, 0.4]	Dropout ratio: 0.3537
	Learning rate: [5 × $10^{-3}$, 0.01]	Learning rate: 0.0057
CNN	Hidden size: [16, 512]	Hidden size: 16
	Dropout ratio: [0.1, 0.4]	Dropout ratio: 0.1507
	Learning rate: [5 × $10^{-3}$, 0.01]	Learning rate: 0.0068
CNN-Att	Hidden size: [16, 512]	Hidden size: 79
	Dropout ratio: [0.1, 0.4]	Dropout ratio: 0.1507
	Learning rate: [5 × $10^{-3}$, 0.01]	Learning rate: 0.0068
DualMode (DTVF)	Hidden size CNN: [16, 512]	Hidden size CNN: 484
	Hidden size LSTM: [64, 512]	Hidden size LSTM: 190
	Num layers: [2, 5]	Num layers: 3
	Dropout ratio CNN: [0.1, 0.4]	Dropout ratio CNN: 0.1099
	Dropout ratio LSTM: [0.1, 0.4]	Dropout ratio LSTM: 0.3072
	Learning rate: [5 × $10^{-3}$, 0.01]	Learning rate: 0.0052

**Table 2 genes-15-01170-t002:** Ablation study of DTVF model.

Model	ACC	SN	PR	FS	SP	AUROC
BiLSTM	0.8220	0.7205	0.9041	0.8019	0.9236	0.9124
CNN	0.8038	0.6979	0.8855	0.7806	0.9097	0.8915
CNN-Att	0.8168	0.7882	0.8361	0.8114	0.8455	0.8941
CNN-Multi	0.8281	0.7830	0.8607	0.8200	0.8732	0.9101
DualMode	0.8455	0.8021	0.8783	0.8385	0.8889	0.9208

**Table 3 genes-15-01170-t003:** Performance comparison of DTVF and existing methods on the test set.

Model	Acc	Sn	Fs	Sp	AUROC
DTVF	0.8455	0.8021	0.8385	0.8889	0.9208
VF-Pred	0.8351	0.82	0.8327	0.85	0.8351
Deep-VF	0.812	0.790	0.807	0.833	
PBVF	0.794	0.774	0.790	0.814	
MP3	0.66	0.536	0.612	0.783	
VirulentPred	0.607	0.641	0.620	0.573	

## Data Availability

All data sources and source code are available from the GitHub repository (http://github.com/niuwa2333/DTVF, accessed on 1 September 2024).

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
