# Peer review of "DTVF: A User-Friendly Tool for Virulence Factor Prediction Based on ProtT5 and Deep Transfer Learning Models"

_genes, 2024, doi:10.3390/genes15091170_

Round 1
Reviewer 1 Report
Comments and Suggestions for Authors
The authors present a thorough research. They exhibit the study’s aims, portray up-to-date literature, explain the methodology, depict the results, highlight the model’s contribution, and underline its limitations.
In the introduction, I advise emphasizing the innovation and originality of the proposed model compared to existing methods.
I recommend revising the conclusions section. The current paragraph should appear in the introduction, while the conclusion should summarize the study’s most important points and contributions.
Comments on the Quality of English LanguageThe paper is eloquent. However, I suggest rewriting the following sentences (I have already amended the writing; of course, you can change the adjustments):
Page 5, lines 132-133: The input vectors’ size is 1024; padding, as is customary in typical convolutional operations, is not required.
Page 5, lines 162-163: The search range of hyperparameters is shown in Table 1.
Page 7, line 196: As the result shown in Table 2, the BiLSTM model achieved an accuracy of 0.8220, sensitivity of 0.7205, precision of 0.9041, F-score of 0.8019, specificity of 0.9236, and an area under the receiver operating characteristic curve of 0.9124.
Page 10, lines 259-260: The pipeline of the workflow of the DTVF User Interface is shown in Figure 6.
Reviewer 2 Report
Comments and Suggestions for Authors
The authors claim to have utilized ProtT5 and Deep Transfer Learning Models to create a user-friendly tool for virulence factor prediction. This study appears practical and could contribute to the field. However, the manuscript lacks detailed descriptions of several steps, making it challenging to assess the validity and correctness of the proposed methods. Major concerns are as follows:
1. Dataset Processing: The authors state that they collected 9,749 virulence factors and 66,982 non-virulence factors from public databases, then applied clustering and CD-HIT for the exclusion of redundant sequences. However, detailed information on how clustering was performed and how CD-HIT was used for redundant sequence exclusion is not clearly presented. It is recommended to provide additional details on these methods.
2. Feature Extraction: The authors claim to use the ProtT5 model to generate feature vectors of 1,024 elements from protein sequences. Since the ProtT5 model is a critical part of the DTVF tool constructed in this study, the theoretical model should be clearly described. Although Figure 2 provides a brief overview of the analysis process, a more detailed explanation of how ProtT5 effectively extracts features is needed. Additionally, the choice of 1,024 as the final output dimension should be justified, and the nature of these 1,024 elements should be explained. Moreover, what information is labeled by these feature vectors?
3. Model Building: The authors use mathematical symbols to describe the theoretical equations for LSTM and CNN algorithms within the dual-channel model. However, the transitions and connections between each symbol are difficult to follow. It is suggested to include the conversion relationships between the layers of the model in Figure 2 or to provide a separate diagram. Additionally, as the output of the DTVF model is a weighted combination of the outputs from the LSTM and CNN modules, please provide a detailed explanation of the form and dimensions of these outputs. How are the weights chosen?
4. Hyperparameter Search: When selecting the best hyperparameters using 10-fold cross-validation, were the hyperparameters for the LSTM and CNN blocks chosen simultaneously or sequentially?
5. Ablation Study: The results show that the accuracy and AUROC of the five models (BiLSTM, CNN, CNN-Att, CNN-Multi, DualMode) do not exhibit significant differences. However, it is suspected that the computational times for these models might vary. Please provide the execution times for each model. Additionally, for Figure 4, the radar chart should have numerical values marked on the axes.
6. Performance Comparison with Existing Models: The authors compare their method with existing methods such as VF-Pred, DeepVF, PBVF, VirulentPred, MP3, and BLAST. They also mention that previous studies lacked precision metrics. Did the authors just extract the results from the literature of these existing tools, or did they use the same data to run each of these tools? For a more detailed comparison, it is recommended to use the same dataset and provide the parameter settings for each tool.
Comments on the Quality of English LanguageNA
Round 2
Reviewer 2 Report
Comments and Suggestions for Authors
The author provided a response to the reviewers' comments; however, they did not make corresponding revisions in the manuscript, and some explanations remain unclear. Below are the major concerns:
1. Dataset Processing: The author claimed to have added the following sentence to the updated version of the manuscript: 'Clustering was performed using the CD-HIT program, which grouped similar sequences together based on a sequence identity threshold of 0.3, and redundant sequences were excluded by selecting representative sequences from each cluster to create a non-redundant dataset.' However, this sentence is not present in the updated version of the manuscript.
2. Feature Extraction:
(Q1) Regarding the suggestion to include a theoretical explanation of the ProtT5 model, which is a key component of the DTVF tool proposed in this study, the author only responded that ProtT5 is similar to other transformer-based models, such as ESM2 and AlphaFold, but did not provide any theoretical explanation of ProtT5. Additionally, since ProtT5 is similar to ESM2 and AlphaFold, what was the reason for choosing ProtT5 in the development of the DTVF tool instead of using other models?
(Q3) It is recommended to clearly list the information regarding amino acid composition, secondary structure propensities, and functional domain locations mentioned by the author in the manuscript.
3. Model Building:
The author claimed that they have provided a sufficient description of the LSTM and CNN algorithms in the manuscript, but the explanations remain unclear to the readers. Since these two algorithms are also important parts of the DTVF tool proposed in this study, it is recommended to provide a clearer explanation. For example, in the LSTM model, the mathematical symbols and formulas used are standard for LSTM, but only the formulas for the layers are provided, without detailed explanations of the relationships and meanings of the symbols, such as i_t, f_t, o_t, etc. This leaves the reader knowing the formulas exist but not understanding their meanings or why the author chose this method. Additionally, the author provided Figure S3 to explain the CNN model, but the figure's meaning is not clearly explained, and it is not referenced in the manuscript. When explaining the CNN model in the manuscript, the author used symbols such as w, x, b, y; it is suggested to use Figure S3 for clarification. Also, Figure S4 is not mentioned in the manuscript.
4. Ablation Study:
(Q3) Numerical values on the axes of the radar chart have not been provided; please ensure they are included.
5. Performance Comparison with Existing Models:
Please include the full explanation from the response in the manuscript. Additionally, the author stated in the response, 'we obtained the evaluation results of these models against the benchmark from relevant literature.' What does this mean? Does it imply that the original studies used this data, and therefore the results were taken from the literature? Please clarify this point.
Comments on the Quality of English LanguageNA
